# Recall and Refine: A Simple but Effective Source-free Open-set Domain Adaptation Framework

**Ismail Nejjar**                                              *ismail.nejjar@epfl.ch*
*EPFL*

**Hao Dong**                                              *hao.dong@ibk.baug.ethz.ch*
*ETH Zürich*

**Olga Fink**                                                  *olga.fink@epfl.ch*
*EPFL*

**Reviewed on OpenReview:** *https://openreview.net/forum?id=HBZoXjUAqV*

## Abstract

Open-set Domain Adaptation (OSDA) aims to adapt a model from a labeled source domain to an unlabeled target domain, where novel classes — also referred to as target-private unknown classes — are present. Source-free Open-set Domain Adaptation (SF-OSDA) methods address OSDA without accessing labeled source data, making them particularly relevant under privacy constraints. However, SF-OSDA presents significant challenges due to distribution shifts and the introduction of novel classes. Existing SF-OSDA methods typically rely on thresholding the prediction entropy of a sample to identify it as either a known or unknown class, but fail to explicitly learn discriminative features for the target-private unknown classes. We propose Recall and Refine (RRDA), a novel SF-OSDA framework designed to address these limitations by explicitly learning features for target-private unknown classes. RRDA employs a two-stage process. First, we enhance the model's capacity to recognize unknown classes by training a target classifier with an additional decision boundary, guided by synthetic samples generated from target domain features. This enables the classifier to effectively separate known and unknown classes. Second, we adapt the entire model to the target domain, addressing both domain shifts and distinguishability to unknown classes. Any off-the-shelf source-free domain adaptation method (e.g. SHOT, AaD) can be seamlessly integrated into our framework at this stage. Extensive experiments on three benchmark datasets demonstrate that RRDA significantly outperforms existing SF-OSDA and OSDA methods. The code is available at `https://github.com/ismailnejjar/RRDA`.

## 1 Introduction

Unsupervised Domain Adaptation (UDA) (Ben-David et al., 2010; Ganin & Lempitsky, 2015; Long et al., 2015) adapts a model from a labeled source domain to an unlabeled target domain (Oza et al., 2023), effectively addressing the issue of domain shift where the source and target distributions differ. UDA strategies typically align feature distributions between domains using metric learning techniques (Long et al., 2015; Kang et al., 2019) or adversarial training (Ganin & Lempitsky, 2015; Tzeng et al., 2017; Luo et al., 2019), and more recently, self-training approaches (Sun et al., 2022; Hoyer et al., 2023; Zhu et al., 2023). Despite their success, most current domain adaptation approaches operate under the assumption of a shared label set between the source and target domains (i.e., $\mathcal{C}_s = \mathcal{C}_t$), referred to as Closed-set Domain Adaptation (Saenko et al., 2010). However, this assumption is often impractical in real-world scenarios.

In contrast, Open-set Domain Adaptation (OSDA) extends the target label space beyond that of the source domain (i.e., $\mathcal{C}_s \subset \mathcal{C}_t$) (Saito et al., 2018; Liu et al., 2019), thereby adding complexity to the DA task.

OSDA aims to align target samples from known classes with those from the source domain while effectively identifying target samples belonging to categories not observed in the source domain, referred to as unknown classes (Panareda Busto & Gall, 2017; Bucci et al., 2020; Jang et al., 2022). Various criteria based on instance-level predictions have been proposed, including entropy-based (Feng et al., 2021; Saito et al., 2020) and confidence-based (Saito & Saenko, 2021; Fu et al., 2020) methods.

Additionally, privacy and legal considerations increasingly limit access to labeled source data for adaptation purposes. To address this, source-free adaptation methods (Fang et al., 2024) have emerged, enabling adaptation without reliance on labeled source data (Kim et al., 2021; Kundu et al., 2020a; Li et al., 2020). In this paper, we focus on Source-free Open-set Domain Adaptation (SF-OSDA), where only a pre-trained source model is available for knowledge transfer, without access to labeled source data. Although some Source-free Domain Adaptation (SF-DA) methods have demonstrated effectiveness in addressing SF-OSDA for classification tasks (Liang et al., 2020; Yang et al., 2022; Wan et al., 2024), semantic segmentation (Choe et al., 2024), and graph applications (Wang et al., 2024b), they primarily focus on the semantics of known classes in the source domain, often overlooking the crucial aspect of novel-class semantics. These methods focus on segregating target samples with low entropy, categorizing them as known classes, and subsequently optimizing specific objectives such as entropy minimization or clustering. In this process, data points associated with known classes are prioritized, while those with high entropy are typically excluded from training, leading to a semantic disparity between the known and unknown classes.

To address these limitations, we propose Recall and Refine (RRDA), a novel SF-OSDA framework that explicitly learns representations for target-private unknown classes by generating synthetic target-domain features. Our approach leverages the key insight that a source classifier trained only on known classes yields high-entropy (highly uncertain) predictions for unknown samples and low-entropy (highly confident) predictions for known ones. We optimize synthetic features to exhibit these entropy characteristics, cluster the resulting high-entropy features into $K'$ pseudo-unknown classes, and refine the source classifier's decision boundaries accordingly to accommodate novel categories. This small change recasts open-set adaptation as a closed-set problem, allowing any off-the-shelf source-free domain adaptation method (e.g., SHOT (Liang et al., 2020), AaD (Yang et al., 2022)) to be seamlessly integrated for joint domain-shift and unknown-class separation. Extensive experiments on three SF-OSDA benchmarks demonstrate that RRDA significantly outperforms existing methods.

## 2 Related Work

**Unsupervised Domain Adaptation (UDA)** aims to adapt a model originally trained on a labeled source domain to perform effectively in an unlabeled target domain. This adaptation process assumes access to data from both the source and target domains during training (Oza et al., 2023). UDA strategies often align feature distributions between domains using metric learning techniques (Long et al., 2015; Kang et al., 2019; Nejjar et al., 2023) or adversarial training across various spaces, including image input space (Murez et al., 2018; Pizzati et al., 2020), feature space (Ganin & Lempitsky, 2015), and output space (Luo et al., 2019; Vu et al., 2019). Additionally, various techniques incorporate pseudo-labeling or self-training algorithms (Sun et al., 2022; Dong et al., 2023; Yue et al., 2024), which generate pseudo-labels for unlabeled samples in the target domain. However, existing approaches assume that label spaces are identical across both domains, limiting their applicability in real-world scenarios.

**Open-set Domain Adaptation (OSDA)** addresses scenarios where the target domain may contain classes not present in the source domain (Panareda Busto & Gall, 2017; Dong et al., 2024a;b; Li et al., 2021). Various approaches have been proposed to tackle this challenge, including assigning target domain images to source categories while discarding unrelated target domain images (Panareda Busto & Gall, 2017), and using adversarial training to separate unknown target samples (Saito et al., 2018; Jang et al., 2022). The Separate to Adapt (STA) approach (Liu et al., 2019) progressively separates unknown and known class samples using a coarse-to-fine weighting mechanism and proposes evaluating OSDA on diverse levels of openness. Rotation-based Open Set (ROS) (Bucci et al., 2020) explores the use of self-supervised tasks such as rotation recognition for unknown class detection. (Jing et al., 2021) projects features to a hyperspherical latent space to reject known samples based on angular distance. Adjustment and Alignment for Unbiased

Open Set Domain Adaptation (ANNA) (Li et al., 2023) addresses semantic-level bias in OSDA by designing Front-Door Adjustment and Decoupled Causal Alignment modules. However, these approaches all assume the availability of labeled source data, which can pose challenges due to privacy concerns in real applications.

**Source-free Domain Adaptation (SFDA)** leverages only a source-trained model and unlabeled target data for adaptation to the target domain. SFDA approaches can be categorized into data-based and model-based methods (Yu et al., 2023). One of the data-driven methods, SHOT, was introduced by Liang et al. (2020). It adapts a pre-trained source model via information maximization with self-supervised pseudo-labeling to implicitly align target domain representations to the source hypothesis. Building on this approach, subsequent works(Chu et al., 2022; Lee et al., 2022; Qu et al., 2022) refine the adaptation through self-training techniques. Other works explore different training procedures. For example, historical Contrastive Learning (HCL) (Huang et al., 2021) compensates for the absence of source data by leveraging historical models and contrasting current and historical embeddings of target samples. Some methods (Yang et al., 2021; 2022) enforce consistency between local neighbors by considering local feature density, with Attract and Disperse (AaD) (Yang et al., 2022) treating SFDA as an unsupervised clustering problem. Additionally, Zhang et al. (2023) explores leveraging source model classifier weights as class prototypes to embed class relationships into a similarity measure for a target sample.

**Source-free Open-set Domain Adaptation (SF-OSDA)** extends SFDA to scenarios where the target domain contains novel classes not present in the source domain. While methods like SHOT (Liang et al., 2020), AaD (Yang et al., 2022), and Uncertainty-guided Source-free Domain Adaptation (U-SFAN) (Roy et al., 2022) have been adapted for SF-OSDA, they primarily focus on the *known* class semantics in the source domain, which can lead to suboptimal handling of target-private unknown classes. Universal Domain Adaptation (UniDA) aims to handle domain shifts and label set differences between source and target domains, encompassing open, partial, and open-partial set scenarios (Liang et al., 2021; Qu et al., 2023; 2024). Recent SF-UniDA methods proposed one-vs-all clustering approaches (Qu et al., 2023) and subspace decomposition (Qu et al., 2024) to separate and identify common and private target classes in a source-free setup. Similarly, Progressive Graph Learning (Luo et al., 2023) decomposes the target hypothesis space into shared and unknown subspaces for SF-OSDA. However, current methods either require specific training for the source model to incorporate the unknown classes (Kundu et al., 2020b;c), which is usually impractical, or rely on thresholding a metric to distinguish known classes from unknown ones during training and inference, making the prediction sensitive to different thresholds.

## 3 Methodology

### 3.1 Preliminary

For SF-OSDA, we are given a source pre-trained model $f_\theta^s$ and an unlabeled target domain with $n_t$ samples, denoted as $\mathcal{D}_t = \{\mathbf{x}_i^t\}_{i=1}^{n_t}$, where $\mathbf{x}_i^t \in \mathcal{X} \subset \mathbb{R}^d$. The target domain follows a distinct data distribution $(P^t \neq P^s)$ from the source domain, reflecting both distribution and label shifts.

Let $\mathcal{C}^s$ and $\mathcal{C}^t \subset \mathcal{Y}$ represent the label sets for the source and target domains, respectively, where $\mathcal{C}^s \subset \mathcal{C}^t$. Both domains share $K$ common classes referred to as *known* classes ($\mathcal{C}_{known}^t = \mathcal{C}^s$). Additionally, target-private classes are grouped as an *unknown* class ($\mathcal{C}_{unk}^t = \mathcal{C}^t \setminus \mathcal{C}^s$). The primary objective of SF-OSDA is to classify both *unknown* and *known* classes, relying exclusively on the target domain data and a pre-trained source model. The pre-trained model can be decomposed as $f_\theta^s = g_\theta^s \circ h_\theta^s$, where $h_\theta^s : \mathbb{R}^d \to \mathbb{R}^D$ is a feature extractor and $g_\theta^s : \mathbb{R}^D \to \mathbb{R}^K$ is the source classifier. Unlike previous works, which freeze the source classifier (e.g. SHOT) during adaptation, we propose training a new target classifier $g_\theta^t$ to explicitly account for target-private unknown classes.

One of the challenges in open-set scenarios is the ability to distinguish known from unknown classes in the target domain. Different approaches have been proposed for distinguishing between known and unknown classes, including hand-crafted thresholding criteria and clustering strategies. However, paradigms such as vendor-to-client (Kundu et al., 2020c) are more effective, as they incorporate an auxiliary out-of-distribution classifier during source training, enabling better handling of unknown classes in the target domain.

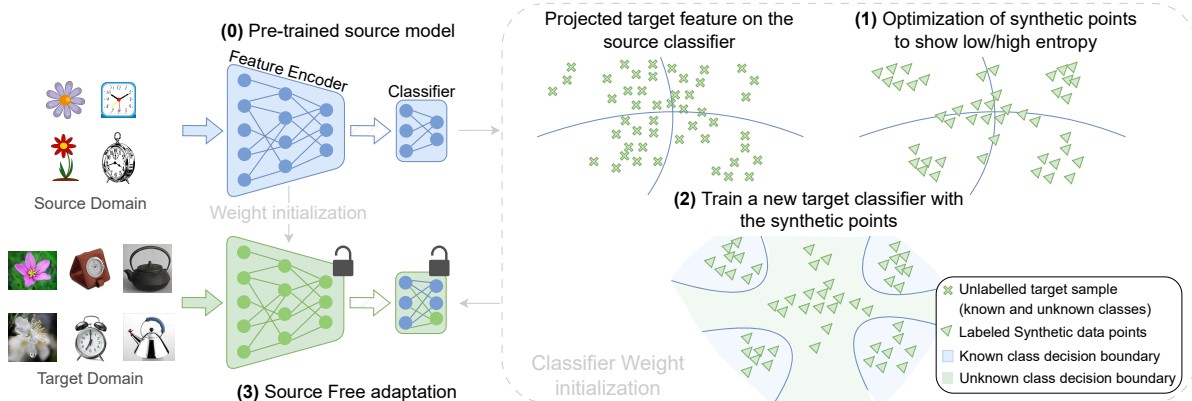

Figure 1: Overview of RRDA for Source-free Open-set Domain Adaptation. Panels (a)–(c) respectively illustrate: (a) target-feature extraction with the frozen source encoder, (b) entropy-guided "recall" of synthetic known/unknown features, (c) "refine" training of a target classifier on these synthetic points, and final adaptation with any closed-set SF-DA objective.

In this paper, we propose a novel approach to address this limitation by adapting the source classifier post hoc to include new decision boundaries for unknown classes. Our method enables the seamless adaptation of any off-the-shelf source pre-trained model to a target domain, even in the presence of novel classes. Motivated by the idea that learning from *unknown* class samples can improve performance in open-set scenarios, *our objective is to simplify adaptation and eliminate the dependency on threshold-based methods during inference.*

## 3.2 RRDA

Our proposed Recall and Refine framework for SF-OSDA consists of three main steps:

1. **Generate synthetic features:** Referring to step (b) in Figure 1, synthetic feature points are generated for both *known* and *unknown* classes. This involves optimizing target feature representations using entropy objectives.

2. **Train a target classifier:** Referring to step (c) in Figure 1, the synthetic feature points are used to train a new target classifier $g_\theta^t$ with extended decision boundaries to accommodate unknown classes.

3. **Adapt the whole model:** The entire model is adapted using any off-the-shelf source-free domain adaptation methods (e.g. SHOT, AaD) on target domain data.

This allows the model to (1) learn the semantics of both known and unknown classes in the target domain, (2) treat OSDA as a simple closed-set scenario, and (3) directly output predictions for unknown classes.

### 3.2.1 Synthetic Data Generation.

The first step of our proposed approach involves generating synthetic features for both *known* and *unknown* classes using the source classifier $g_\theta^s$. Specifically, we optimize the target feature representation $\mathbf{z}^t = h_\theta^s(x^t)$ to generate synthetic samples that exhibit low entropy for *known* classes and high entropy for the *unknown* class. We denote these optimized synthetic features as $\mathbf{z}_{known}^{*t}$ and $\mathbf{z}_{unk}^{*t}$. The resulting unknown features are clustered into $K'$ groups, and a new target classifier $g_\theta^t$ is introduced with $K + K'$ classes.

In this section, we describe the process for obtaining feature representations for both *known* and *unknown* classes. We use standard gradient descent optimization to generate the desired feature representations.

**Synthetic *Unknown* Classes Generation:** To effectively identify points near the source classifier's decision boundary, we seek synthetic features $\mathbf{z}_{unk}^{*t}$ that maximize the entropy while ensuring diverse feature

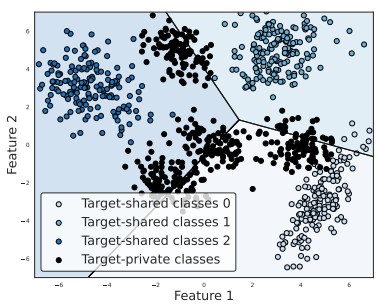
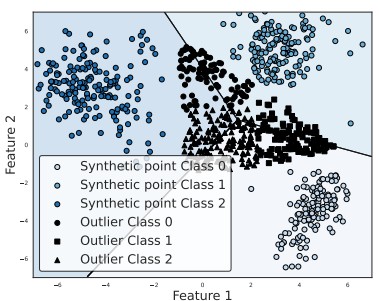
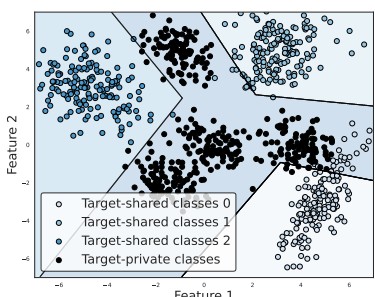

(a) Unlabeled target domain projected onto the decision boundary of the source classifier.

(b) Optimized synthetic points for known and unknown classes.

(c) Unlabeled target domain projected onto the new decision boundary of the target classifier.

Figure 2: Visualization of the synthetic data generation process and the resulting target classifier boundary on a toy example with $K = K' = 3$ classes.

representations, thereby reducing the risk of collapsing to a single point in feature space. To prevent feature collapse, we add a hinge-style variance regularizer $\max\{0, 1 - \sqrt{\mathrm{Var}(\mathbf{z}^t)}\}$ computed over the entire set of synthetic features, explicitly enforcing $\mathrm{Var}(\mathbf{z}^t) \geq 1$ and encouraging global diversity, similar to Bardes et al. (2022). Specifically, we initialize the optimization with a noisy version of the original features $\mathbf{z}^t$ by adding Gaussian noise drawn from $\mathcal{N}(0, 1.0)$ to the features extracted by the source-pretrained encoder $h_\theta^s$ before any optimization.

$$\min_{\mathbf{z}^t} \quad -H(\sigma(g_\theta^s(\mathbf{z}^t))) + \lambda \cdot \max\{0, 1 - \sqrt{\mathrm{Var}(\mathbf{z}^t)}\}, \tag{1}$$

where $H(p) = -\sum_{k=1}^{K} p_k \log(p_k)$ represents the entropy, and $\sigma$ is the softmax activation function, and $\lambda$ was set to 1 for all the experiments. After optimization, we select feature points with high entropy as unknown class candidates. Specifically, only points satisfying $H(\sigma(g_\theta^s(\mathbf{z}^{*t}))) > 0.75 \cdot \log(K)$ are considered as $\mathbf{z}_{unk}^{*t}$. The threshold value 0.75 is analyzed in the Ablation section. The selected high-entropy features $\mathbf{z}_{unk}^{*t}$ are then clustered into $K'$ distinct unknown classes using K-means clustering. Each resulting cluster is assigned a unique label in the range $y_{unk}^{*t} \in \{K+1, \ldots, K+K'\}$, effectively extending the label space beyond the original $K$ known classes. This approach is motivated by the observation in the literature (Lampert et al., 2009) that it is possible to generate meaningful semantics for novel classes using known classes.

**Synthetic *Known* Classes Generation:** A similar optimization approach is employed to generate synthetic data points for the *known* classes. The optimization is performed iteratively $K$ times, once for each known class $k$ (where $k \in \{1, ..., K\}$). The objective is to minimize the cross-entropy for each class directly from $\mathbf{z}^t$. The optimization problem for generating a sample for class $k$ is defined as:

$$\min_{\mathbf{z}^t} \quad \mathcal{L}_{CE}(g_\theta^s(\mathbf{z}^t), k) + \lambda \cdot \max\{0, 1 - \sqrt{\mathrm{Var}(\mathbf{z}^t)}\}, \tag{2}$$

where $\mathcal{L}_{CE}$ is the cross-entropy loss that encourages the model to predict class $k$, and $\lambda$ controls the regularization term, set to 1 in all experiments. After optimization, we select feature points with low cross-entropy loss as known class candidates. Specifically, only points satisfying $\mathcal{L}_{CE}(g_\theta^s(\mathbf{z}^{*t}), I_k) < 0.25 \cdot \log(K)$ are considered as $\mathbf{z}_{known}^{*t}$. This threshold ensures the synthetic features have high confidence for their assigned class. The specific value 0.25 is analyzed in the Ablation section.

Each synthetic feature $\mathbf{z}_{known}^{*t}$ is assigned the label $y_{known}^{*t} = k$, forming the pairs $(\mathbf{z}_{known}^{*t}, y_k^{*t})$. These synthetic data points and their corresponding pseudo-labels are then used to train the target classifier. By iteratively generating feature points for each known class, our method enhances the decision boundaries without requiring access to the original source data or labels.

### 3.2.2 Target Classifier Training.

In the second step, we instantiate a new target classifier $g_\theta^t : \mathbb{R}^D \to \mathbb{R}^{K+K'}$ that accommodates both known and unknown classes. The weights for the known classes $g_{\theta_{[1:K]}}^t$ are initialized using the source classifier's weights $g_\theta^s$, while the weights for the unknown classes $g_{\theta_{[K+1:K+K']}}^t$ are randomly initialized.

The target classifier is trained using the synthetic features and their corresponding labels from the previous step. For known classes, we use pairs $(\mathbf{z}_{\text{known}}^{*t}, y_{\text{known}}^{*t})$ where $y_{\text{known}}^{*t} \in \{1, \ldots, K\}$, and for unknown classes, we use pairs $(\mathbf{z}_{\text{unk}}^{*t}, y_{\text{unk}}^{*t})$ where $y_{\text{unk}}^{*t} \in \{K+1, \ldots, K+K'\}$.

We optimize the classifier parameters through standard cross-entropy minimization:

$$\min_\theta \; \mathcal{L}_{\text{CE}}(g_\theta^t(\mathbf{z}^{*t}), y^{*t}), \tag{3}$$

where $\mathbf{z}^{*t}$ represents all synthetic features and $y^{*t}$ their corresponding labels. Figure 2 illustrates the resulting decision boundaries that separate both known and unknown classes.

### 3.2.3 Target Domain Adaptation.

Any source-free unsupervised domain adaptation method (originally designed for closed-set scenarios) can be integrated into our approach to address open-set scenarios, provided it incorporates a diversity loss or a similar mechanism to facilitate self-learning of unknown classes. To empirically validate this hypothesis, we consider SHOT (Liang et al., 2020) and AaD (Yang et al., 2022), using their respective training objectives for adaptation. SHOT (Liang et al., 2020) employs information maximization and self-supervised pseudo-labeling to adapt the source model to the target domain. Its objective function can be expressed as:

$$\mathcal{L}_{\text{shot}} = -\frac{\lambda_{ent}}{n_t} \sum_{i=1}^{n_t} \sum_{k=1}^{K+K'} p_{k,i} \log p_{k,i} + \lambda_{\text{div}} \cdot \sum_{k=1}^{K+K'} \bar{p}_k \log \bar{p}_k + \lambda_{\text{ps}} \cdot \mathcal{L}_{\text{pseudo}}, \tag{4}$$

where $\bar{p}_k = \frac{1}{n_t} \sum_{i=1}^{n_t} p_k(x_i; \theta)$, and $\mathcal{L}_{\text{pseudo}}$ is the pseudo-labeling loss function from Liang et al. (2020). During adaptation, only the feature encoder is updated while the classifier remains frozen. AaD (Yang et al., 2022) leverages local consistency and global dispersion. The objective function for feature $i$ is formulated as:

$$\mathcal{L}_{\text{AaD},i} = -\sum_{j \in \mathcal{C}_i} p_i^T p_j + \lambda \sum_{m \in \mathcal{B}_i} p_i^T p_m, \tag{5}$$

where $\mathcal{C}_i$ represents the local neighborhood of feature $i$ and $\mathcal{B}_i$ is the mini-batch feature not in $\mathcal{C}_i$. Unlike SHOT, AaD updates the entire model weights during adaptation.

## 4 Experiments

### 4.1 Experimental Setup

**Datasets: Office-Home** (Venkateswara et al., 2017) comprises 65 labeled image categories from four distinct domains: Art (Ar), Clipart (Cl), Product (Pr), and Real World (Rw). We designate the first 25 alphabetically ordered categories as known classes, with the remaining 40 as unknown. **Office-31** (Saenko et al., 2010) consists of 31 classes across three domains: Amazon (A), Dslr (D), and Webcam (W). We assign the first 10 as known and the last 10 classes as unknown. **VisDA** (Peng et al., 2017) has 12 categories across two domains: Real (R) and Synthetic (S). The first 6 classes are categorized as known and the remaining 6 as unknown.

**Evaluation Metrics:** To assess model performance, we adopt standard evaluation metrics widely used in previous OSDA studies (Bucci et al., 2020; Liu et al., 2019; Li et al., 2023). The Harmonic Open-set (HOS) accuracy balances performance on known and unknown classes and can be calculated as $HOS = \frac{2 \times OS^* \times UNK}{OS^* + UNK}$, where $OS^*$ represents the accuracy of known classes, and $UNK$ denotes the accuracy of

Table 1: HOS (%) results on Office-31 (ResNet-50). SF denotes source-free methods. AaD-O and SHOT-O are the adapted open-set methods of AaD and SHOT. RRDA uses standard AaD and SHOT versions.

| Methods | SF | Office-31 | | | | | | |
|---|---|---|---|---|---|---|---|---|
| | | A2D | A2W | D2A | D2W | W2A | W2D | Avg |
| CMU | ✗ | 52.6 | 55.7 | 76.5 | 75.9 | 65.8 | 64.7 | 65.2 |
| DANCE | ✗ | 84.9 | 78.8 | 79.1 | 78.8 | 68.3 | 78.8 | 79.8 |
| OSLPP | ✗ | 91.5 | 89.0 | 79.3 | 92.3 | 78.7 | 93.6 | 87.4 |
| GATE | ✗ | 88.4 | 86.5 | 84.2 | 95.0 | 86.1 | 96.7 | 89.5 |
| ANNA | ✗ | 83.8 | 85.5 | 82.5 | 99.5 | 81.6 | 98.4 | 88.6 |
| Source-only | ✓ | 78.2 | 72.1 | 44.2 | 82.2 | 52.1 | 88.8 | 69.6 |
| UMAD | ✓ | 88.5 | 84.4 | 86.8 | 95.0 | 88.2 | 95.9 | 89.8 |
| LEAD | ✓ | 84.9 | 85.1 | 90.9 | 94.8 | 90.3 | 96.5 | 90.3 |
| GLC | ✓ | 82.6 | 74.6 | 92.6 | 96.0 | 91.8 | 96.1 | 89.0 |
| AaD-O | ✓ | 82.3 | 79.0 | 84.3 | 93.1 | 84.8 | 95.0 | 86.4 |
| AaD + RRDA | ✓ | 91.1 | 94.3 | 94.1 | 96.6 | 94.0 | 96.2 | **94.4** |
| | | +8.8 | +15.3 | +9.8 | +3.5 | +9.2 | +1.2 | **+8.0** |
| SHOT-O | ✓ | 89.5 | 83.0 | 85.9 | 91.4 | 84.0 | 95.2 | 88.2 |
| SHOT+ RRDA | ✓ | 90.0 | 92.2 | 92.6 | 98.2 | 91.6 | 98.2 | **93.8** |
| | | +0.5 | +9.2 | +6.7 | +6.8 | +7.6 | +3.0 | **+5.6** |

unknown classes. The HOS metric provides a comprehensive measure, by equally weighting the model's ability to classify known classes and detect unknown classes.

**Implementation Details:** All experiments are conducted on a single A100 GPU using PyTorch. For synthetic data generation, we employ the Adam optimizer with a learning rate of 0.001 for 1000 steps, for both known ($\mathbf{z}_k^{*t}$) and unknown ($\mathbf{z}_{unk}^{*t}$) classes. In all main experiments, we set $K' = K$. To maintain class balance, we cap the sample size at 1000 for known classes in Office-Home and Office-31, and $10,000$ for VisDA. The target classifier is trained for 50 epochs using SGD with a learning rate of 0.01, momentum of 0.9, weight decay of 0.001, and a fixed batch size of 128. During target model adaptation, we use SGD with momentum 0.9, a batch size of 64, and train for 50 epochs. The learning rate is set to 0.001 for Office-31 and Office-Home, and 0.0001 for VisDA, when using Resnet-50 (He et al., 2016) as the backbone. When using ViT-B (Wu et al., 2020) as the backbone, we set the learning rate to 0.0001 for all experiments. For SHOT, we freeze the target classifier and train only the feature extractor and backbone. For AaD, all model parameters are trained, with the feature extractor's learning rate set to 10 times lower. During inference, samples belonging to the new $K'$ classes are considered unknown target samples. $\lambda_{\text{ps}}$ was set to 0.1, 0.3, and 0.4 for Office-Home, Office-31, and VisDA respectively. The hyper-parameters $\lambda_{\text{div}}$ and $\lambda$ were set to 1 and $\lambda_{ent}$ was set to 0.5 in all our experiments.

During inference, samples assigned to any of the new $K'$ classes are treated as unknown target samples. Following the standard open-set protocol, predictions in the range $[1, K]$ correspond to known classes, while predictions in $[K + 1, K + K']$ are aggregated into a single $K + 1$ class, representing the unknown class.

**Baselines:** We compare our method against open-set domain adaptation approaches, including both non-source-free and source-free methods. The non-source-free methods include OSBP (Saito et al., 2018), CMU (Fu et al., 2020), STA (Liu et al., 2019), DANCE (Saito et al., 2020), GATE (Chen et al., 2022), ANNA (Li et al., 2023), and OSLPP (Wang et al., 2024a). For source-free methods, we consider UMAD (Liang et al., 2021), GLC (Qu et al., 2023), SF-PGL (Luo et al., 2023), and LEAD (Qu et al., 2024).

## 4.2 Experimental Results

From Table 1 to 3, we compare our method against state-of-the-art (SOTA) OSDA methods in both source-free and non-source-free setups. We include non-SF methods to provide a comprehensive performance benchmark, despite our focus on SF scenarios. We use RRDA alongside AaD and SHOT (vanilla methods for closed-set scenarios), as well as their open-set variants denoted as AaD-O and SHOT-O that rely on entropy-

Table 2: HOS (%) results on Office-Home (ResNet-50 and ViT). $|C_s| = 25$, $|C_t| = 65$. SF denotes source-free.

| Methods | SF | Office-Home | | | | | | | | | | | | |
|---|---|---|---|---|---|---|---|---|---|---|---|---|---|---|
| | | Ar2Cl | Ar2Pr | Ar2Rw | Cl2Ar | Cl2Pr | Cl2Rw | Pr2Ar | Pr2Cl | Pr2Rw | Rw2Ar | Rw2Cl | Rw2Pr | **Avg** |
| CMU | ✗ | 55.0 | 57.0 | 59.0 | 59.3 | 58.2 | 60.6 | 59.2 | 51.3 | 61.2 | 61.9 | 53.5 | 55.3 | 57.6 |
| DANCE | ✗ | 6.5 | 9.0 | 9.9 | 20.4 | 10.1 | 9.2 | 28.1 | 15.8 | 12.6 | 14.2 | 7.9 | 13.7 | 12.9 |
| OSLPP | ✗ | 61.0 | 72.8 | 74.3 | 60.9 | 66.9 | 70.4 | 63.6 | 59.3 | 74.0 | 67.2 | 59.0 | 74.4 | 67.0 |
| GATE | ✗ | 63.8 | 70.5 | 75.8 | 66.4 | 67.9 | 71.7 | 67.3 | 61.3 | 76.0 | 70.4 | 61.8 | 75.4 | 69.0 |
| ANNA | ✗ | 69.0 | 73.7 | 76.8 | 64.7 | 68.6 | 73.0 | 66.5 | 63.1 | 76.6 | 71.3 | 65.7 | 78.7 | 70.7 |
| Source-only | ✓ | 46.1 | 63.3 | 72.9 | 42.8 | 54.0 | 58.7 | 47.8 | 36.1 | 66.2 | 60.8 | 45.3 | 68.2 | 55.2 |
| UMAD | ✓ | 59.2 | 71.8 | 76.6 | 63.5 | 69.0 | 71.9 | 62.5 | 54.6 | 72.8 | 66.5 | 57.9 | 70.7 | 66.4 |
| LEAD | ✓ | 60.7 | 70.8 | 76.5 | 61.0 | 68.6 | 70.8 | 65.3 | 59.8 | 74.2 | 64.8 | 57.7 | 75.6 | 67.2 |
| GLC | ✓ | 65.3 | 74.2 | 79.0 | 60.4 | 71.6 | 74.7 | 63.7 | 63.2 | 75.8 | 67.1 | 64.3 | 77.8 | 69.8 |
| AaD-O | ✓ | 58.0 | 68.2 | 75.4 | 58.8 | 65.7 | 69.0 | 54.6 | 52.9 | 72.3 | 65.8 | 56.3 | 72.2 | 64.1 |
| AaD + RRDA | ✓ | 61.7 | 72.8 | 73.5 | 59.0 | 74.9 | 69.9 | 59.5 | 58.3 | 71.2 | 64.5 | 64.8 | 73.2 | 66.9 |
| | | +3.7 | +4.6 | -1.9 | +0.2 | +9.2 | +0.9 | +4.9 | +5.4 | -1.1 | -1.3 | +7.7 | +1.0 | **+2.8** |
| SHOT-O | ✓ | 57.2 | 65.4 | 69.9 | 58.1 | 62.6 | 64.3 | 60.5 | 52.8 | 71.1 | 64.4 | 53.5 | 40.6 | 61.9 |
| SHOT + RRDA | ✓ | 64.6 | 74.2 | 77.2 | 63.1 | 71.4 | 71.3 | 67.7 | 59.1 | 76.7 | 70.2 | 67.4 | 76.7 | **70.0** |
| | | +7.4 | +8.8 | +7.3 | +5.0 | +8.8 | +7.0 | +7.2 | +6.3 | +5.6 | +5.8 | +13.9 | +36.1 | **+8.1** |
| Source-only | ✓ | 57.1 | 69.5 | 79.9 | 50.2 | 62.5 | 66.0 | 52.2 | 45.7 | 75.1 | 69.3 | 56.4 | 73.7 | 63.1 |
| LEAD | ✓ | 58.6 | 74.7 | 82.7 | 58.9 | 74.6 | 74.3 | 59.0 | 47.1 | 78.3 | 71.9 | 58.7 | 77.4 | 68.0 |
| AaD-O | ✓ | 57.8 | 74.9 | 82.7 | 53.9 | 68.6 | 70.8 | 52.5 | 45.8 | 76.8 | 70.6 | 58.2 | 77.7 | 65.9 |
| AaD + RRDA | ✓ | 67.4 | 77.2 | 81.2 | 71.4 | 71.5 | 76.1 | 73.9 | 63.8 | 78.5 | 74.8 | 67.5 | 75.0 | **73.2** |
| | | +9.6 | +1.5 | -1.5 | +17.5 | +2.9 | +5.3 | +21.4 | +18.0 | +1.7 | +4.2 | +9.3 | -2.7 | **+7.3** |
| SHOT-O | ✓ | 63.6 | 73.5 | 81.7 | 66.7 | 69.8 | 75.5 | 66.5 | 56.2 | 79.0 | 73.5 | 62.6 | 74.6 | 70.3 |
| SHOT + RRDA | ✓ | 68.9 | 75.0 | 81.4 | 71.2 | 73.8 | 73.6 | 71.9 | 60.4 | 79.2 | 76.5 | 66.2 | 77.5 | **73.0** |
| | | +5.3 | +1.5 | -0.3 | +4.5 | +4.0 | -2.1 | +5.4 | +4.2 | +0.2 | +3.0 | +3.6 | +2.9 | **+2.7** |

*(ResNet-50 — first block; ViT — second block)*

thresholding during training and inference. Results for comparison methods are sourced from (Qu et al., 2024; Li et al., 2023), and the mean HOS is reported.

**Office-31.** Table 1 presents results on the Office-31 dataset, where RRDA demonstrates significant improvements over threshold-based methods. AaD+RRDA achieves an average HOS of 94.4%, which is an 8.0% increase over AaD-O. Similarly, SHOT+RRDA reaches 93.8%, representing a 5.6% improvement over SHOT-O. These results surpass all compared source-free and non-source-free SOTA methods.

**Office-Home.** On the Office-Home dataset (Table 2), RRDA consistently enhances the performance of both AaD-O and SHOT-O across most domain adaptation tasks. AaD+RRDA and SHOT+RRDA show average HOS improvements of +2.8% and +8.1% respectively. SHOT+RRDA achieves a competitive 70.0% average HOS, outperforming most methods, including both source-free and non-source-free approaches, while falling just slightly short of ANNA a non-source-free adaptation method. A similar observation can be made when using ViT as a backbone, where RRDA consistently improves previous methods AaD+RRDA and SHOT+RRDA show average HOS improvements of +7.3% and +2.7% respectively, and surpass the other baselines.

**VisDA.** On the challenging VisDA dataset (Table 3), RRDA continues to demonstrate its effectiveness. AaD+RRDA improves upon AaD-O by +6.0% in HOS, significantly improving unknown sample recognition and overall class accuracy. Significant improvements are observed in classes such as "Bus" (+15.9%) and "Truck" (+20.4%). Similar improvements can be observed when applying RRDA to SHOT with an improvement of +22.7% in HOS. We observe that both methods improve over the same class and degrade the performances of the "car" and "motorcycles" classes.

Table 3: Accuracy for each class (%) and HOS (%) results on VisDA (ResNet-50 and ViT), with $|C_s| = 6$, $|C_t| = 12$.

| Methods | VisDA | | | | | | | | |
|---|---|---|---|---|---|---|---|---|---|
| | Bic | Bus | Car | Mot | Tra | Tru | UNK | HOS | |
| OSBP | 35.6 | 59.8 | 48.3 | 76.8 | 55.5 | 29.8 | 81.7 | 62.7 | |
| STA | 50.1 | 69.1 | 59.7 | 85.7 | 84.7 | 25.1 | 82.4 | 71.0 | |
| Source-only | 16.3 | 7.9 | 24.9 | 48.0 | 6.1 | 0.0 | 72.7 | 27.9 | |
| LEAD | 83.5 | 65.2 | 57.7 | 35.7 | 82.1 | 79.5 | 82.7 | 74.2 | ResNet-50 |
| SF-PGL | 91.5 | 90.1 | 74.1 | 90.3 | 81.9 | 74.8 | 72.0 | 77.4 | |
| AaD-O | 86.8 | 69.8 | 51.5 | 38.7 | 84.3 | 26.0 | 65.5 | 62.4 | |
| AaD + RRDA | 96.0 | 85.7 | 34.5 | 37.6 | 92.2 | 46.4 | 71.7 | 68.4 | |
| | +9.2 | +15.9 | -17.0 | -1.1 | +7.9 | +20.4 | +6.2 | **+6.0** | |
| Shot-O | 82.1 | 67.0 | 78.6 | 57.3 | 72.2 | 17.9 | 50.7 | 56.0 | |
| Shot + RRDA | 88.6 | 82.2 | 66.8 | 47.3 | 87.2 | 74.3 | **83.5** | **78.7** | |
| | +6.5 | +15.2 | -11.8 | -10.0 | +15.0 | +56.4 | +32.8 | **+22.7** | |
| Source-only | 62.3 | 17.9 | 17.7 | 50.7 | 0.0 | 0.6 | 90.8 | 39.1 | |
| LEAD | 87.6 | 65.3 | 49.8 | 30.5 | 70.9 | 54.4 | 98.2 | 74.3 | |
| AaD-O | 87.9 | 77.6 | 47.7 | 36.8 | 61.2 | 16.6 | 67.6 | 60.4 | |
| AaD + RRDA | 98.2 | 91.1 | 84.8 | 39.4 | 94.2 | 97.5 | 81.7 | **82.9** | ViT |
| | +10.3 | +13.5 | +37.1 | +2.6 | +33.0 | +80.9 | +14.1 | +22.5 | |
| Shot-O | 96.7 | 77.0 | 80.4 | 75.9 | 3.0 | 4.7 | 80.1 | 66.1 | |
| Shot + RRDA | 95.4 | 84.8 | 74.1 | 48.7 | 85.1 | 82.3 | 79.3 | **78.9** | |
| | -1.3 | +7.8 | -6.3 | -27.2 | +82.1 | +77.6 | -0.8 | + 12.8 | |

Table 4: Impact of the variance regularization term on Office-Home dataset using SHOT+RRDA (HOS %).

| Method | Ar2Cl | Ar2Pr | Ar2Rw | Cl2Ar | Cl2Pr | Cl2Rw | Pr2Ar | Pr2Cl | Pr2Rw | Rw2Ar | Rw2Cl | Rw2Pr | **Avg** |
|---|---|---|---|---|---|---|---|---|---|---|---|---|---|
| w/o variance term | 59.9 | 70.1 | 68.3 | 60.6 | 67.5 | 68.0 | 58.8 | 58.4 | 75.7 | 56.5 | 55.9 | 77.2 | 64.7 |
| w/ variance term | **64.6** | **74.2** | **77.2** | **63.1** | **71.4** | **71.3** | **67.7** | **59.1** | **76.7** | **70.2** | **67.4** | 76.7 | **70.0** |
| | +4.7 | +4.1 | +8.9 | +2.5 | +3.9 | +3.3 | +8.9 | +0.7 | +1.0 | +13.7 | +11.5 | -0.5 | **+5.3** |

These results demonstrate RRDA's consistent superiority across various domain adaptation scenarios. Our method significantly improves existing SF-OSDA techniques, as evidenced by the consistent performance gains across all datasets. The key advantage of RRDA lies in its novel approach to handling unknown classes. Unlike previous methods that rely on thresholding and discard unknown class data during adaptation, RRDA actively learns the semantics of unknown classes through our adaptive target classifier, which evolves to accommodate the unknown class distribution. Furthermore, the consistent performance gains with both AaD and SHOT demonstrate RRDA's versatility. These results underscore the importance of explicitly modeling unknown classes in open-set domain adaptation, rather than treating them as outliers to be discarded.

## 5    Ablation Study and Sensitivity Analysis

**Impact of Variance Term.** To evaluate the importance of the variance regularization term in our optimization objective, we conducted ablation experiments on the Office-Home dataset using SHOT+RRDA. As shown in Table 4, incorporating the variance term yields an average improvement of +5.3% in HOS across all transfer tasks. The variance term proves especially effective for challenging transfer tasks, with substantial improvements on the most difficult transfers: Rw→Ar (+13.7%) and Rw→Cl (+11.5%). This confirms the importance of promoting feature diversity during synthetic sample generation, particularly when adapting from complex source domains like Real World.

Table 5: Sensitivity to entropy thresholds on Office–31 using SHOT+RRDA.

| Metrics | Threshold (low/high) | | | | | |
|---|---|---|---|---|---|---|
| | 0.10/0.90 | 0.20/0.80 | **0.25/0.75** | 0.30/0.70 | 0.40/0.60 | 0.50/0.50 |
| HOS (%) | 70.5 | 71.5 | **72.6** | 72.0 | 72.3 | 72.1 |
| OS* (%) | 88.7 | 89.4 | **90.0** | 89.9 | **90.0** | 89.0 |
| UNK (%) | 59.6 | 61.1 | **62.3** | 61.4 | 61.6 | 61.9 |

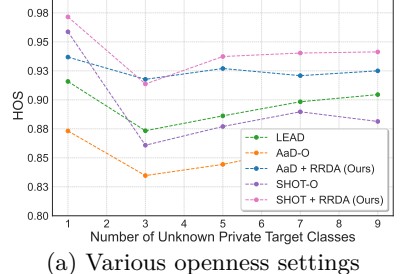

(a) Various openness settings

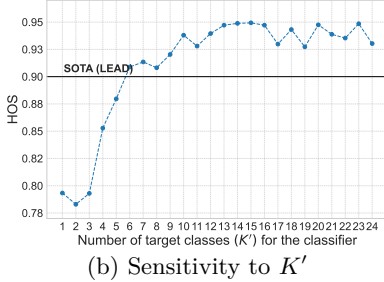

(b) Sensitivity to $K'$

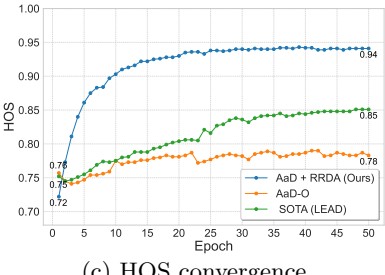

(c) HOS convergence

Figure 3: Sensitivity analysis on Office-31. (a) Adaptation performance across different openness levels (average across all transfer tasks). (b) Sensitivity to $K'$ target classifier classes (average across all transfer tasks). (c) HOS curves for the A2W task.

**Threshold Sensitivity Analysis.** We conducted a comprehensive analysis of entropy threshold sensitivity on the Office-31 dataset using SHOT+RRDA. These thresholds control the selection criteria for synthetic features, with low/high entropy thresholds determining the inclusion boundaries for known/unknown class candidates respectively. As shown in Table 5, our approach maintains consistent performance across various threshold configurations, with the 0.25/0.75 setting achieving optimal balance with the highest HOS of 72.6%. We maintain this threshold setting throughout all experiments to ensure consistency across datasets.

**Varying Unknown Classes.** We investigated the robustness of our framework against an increased number of unknown private classes, which complicates the distinction between known and unknown classes. We compared our method to LEAD, SHOT-O, and AaD-O on the Office-31 dataset. As shown in Figure 3a, our RRDA method in combination with SHOT and AaD achieves stable results and consistently outperforms existing approaches. For consistency with our main results, we kept $K'$ fixed at 10.

**Sensitivity to K'.** Figure 3b shows adaptation performance for different $K'$ values of the target classifier on Office-31 dataset. The performance improves as $K'$ increases, validating the benefit of inheriting class separability knowledge, before eventually reaching a plateau. In fact, $K' = 15$ yields the best results. For the main experiments, we reported Office-31 results using $K' = K = 10$.

**Training Stability.** Figure 3c illustrates the training curves for the A2W task on the Office-31 dataset. Our method shows consistent HOS improvement on the test set, with steadily increasing before plateauing. In contrast, AaD-O exhibits unstable training, with noticeable performance fluctuations throughout the training process.

**Computational Overhead.** We measured runtime comparisons on the Office-31 A2D task as shown in Table 6. As shown, RRDA introduces only a small computational overhead (approximately 31 seconds, or 13% increase) while achieving significant performance gains (+5.6% HOS improvement for SHOT+RRDA over SHOT-O on Office-31). The additional computation time primarily comes from the synthetic data generation and target classifier training stages, which require only a small fraction of the total adaptation time.

Table 6: Runtime comparison of different methods on Office-31 A2D task.

| Method | Runtime |
|---|---|
| LEAD | 3m59s |
| SHOT | 4m00s |
| SHOT+RRDA | 4m31s |
| AaD | 4m01s |
| AaD+RRDA | 4m35s |

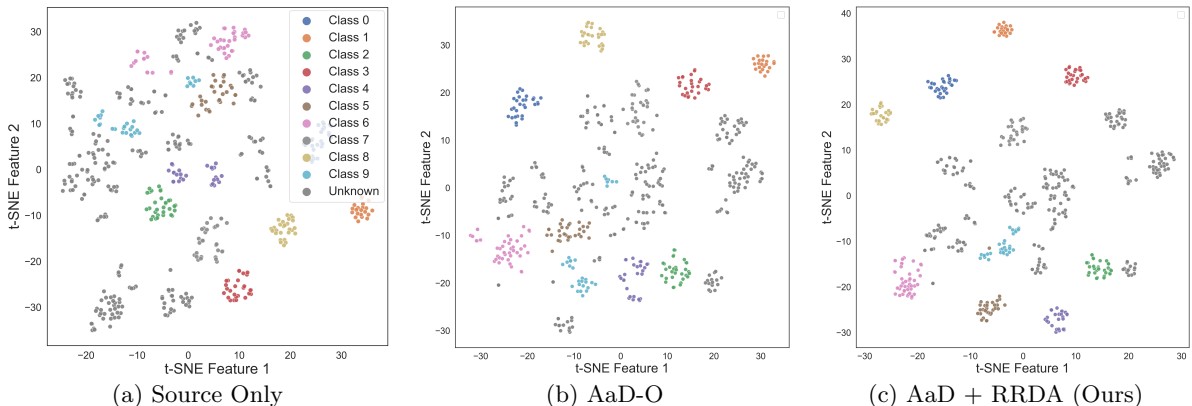

Figure 4: T-SNE visualization of the pre-classifier feature space for the A2W task on the Office-31 dataset.

**Feature Space Visualization.** Figure 4 shows t-SNE embeddings of pre-classifier features for the source-only model, AaD-O, and our method on the A2W task on the Office-31 dataset. The source-only model (Figure 4a) exhibits well-separated known class clusters but mixes unknown samples with known classes. AaD-O (Figure 4b) slightly improves known-unknown separation, but class overlap remains. Our method (Figure 4c) achieves superior separation of known and unknown classes, maintaining tight, well-defined known class clusters while isolating unknown samples. This demonstrates our method's effectiveness in inheriting class separability during adaptation.

## 6 Conclusion

In this work, we introduce **R**ecall and **R**efine for **D**omain **A**daptation (RRDA), a simple but effective framework for SF-OSDA. RRDA enables the successful adaptation of off-the-shelf source pre-trained models to target domains, effectively addressing both distribution and category shift problems. RRDA achieves this by introducing a new target classifier that aids in classifying and learning the semantics of both known and unknown classes. This approach enables the direct use of source-free adaptation methods designed for closed-set scenarios in open-set contexts. Extensive experiments on three challenging benchmarks demonstrate that RRDA significantly outperforms existing SF-OSDA methods and even surpasses OSDA methods that have access to the source domain. Future work could explore its potential for continuous adaptation in the setup where new classes appear over time.

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

## A    Appendix

This supplementary material contains additional information and experiments to support our main paper on source-free open-set domain adaptation. Specifically, it includes additional experiments on foundation model integration, detailed ablation studies on target classifier initialization, and a future direction section.

## B    Ablation Study

### B.1    Target Classifier Initialization

We present an ablation study comparing the impact of known class initialization on the final performance of RRDA on the Office-31 dataset.

In our approach, we first generate synthetic points for both known and unknown classes. These synthetic points are then used to train the target classifier before performing the adaptation. Our main paper uses source initialization for the known classes in the target classifier before this training process.

As shown in Table 7, source initialization of the known classes yields a consistent improvement over random initialization. Importantly, both SHOT+RRDA variants (random and source initialization of the target classifier) significantly outperform the baseline SHOT-O method, with average improvements of 5.0% and 5.6%, respectively.

These findings demonstrate that our synthetic data generation process is robust and effective. Even when the target classifier is randomly initialized before training with the synthetic points, our method achieves strong performance. This underscores the quality of the generated synthetic points for both known and unknown classes.

| Methods | Target Classifier | Office-31 | | | | | | |
|---|---|---|---|---|---|---|---|---|
| | | A2D | A2W | D2A | D2W | W2A | W2D | Avg |
| SHOT-O | | 89.5 | 83.0 | 85.9 | 91.4 | 84.0 | 95.2 | 88.2 |
| SHOT+ RRDA | Random Initialization | 90.0 | 91.5 | 92.4 | 98.0 | 89.0 | 98.4 | 93.2 |
| SHOT+ RRDA | Source Initialization | 90.0 | 92.2 | 92.6 | 98.2 | 91.6 | 98.2 | **93.8** |

Table 7: HOS (%) results on Office-31 (ResNet-50). Comparison of the impact of the target classifier initialization on the final performance.

### B.2    Foundation Model Integration

Recent trends in domain adaptation have focused on leveraging foundation models to tackle the domain shift problem Tang et al. (2024); Zara et al. (2023); Singha et al. (2023). To demonstrate RRDA's flexibility and compatibility with modern vision-language models, we conducted additional experiments integrating our approach with CLIP Radford et al. (2021).

For our experimental setup, we used the Office-Home dataset. We didn't finetune the model, and considered 25 shared classes. Our CLIP configuration included both zero-shot classification using text embeddings of "a photo of [CLASS]" prompts, and our method initialized with CLIP-ViT-B/32 image encoder. We augmented this with entropy-based unknown detection (threshold = $0.5 \times \log(25)$), as we considered only the known classes as prompts.

For our approach, we leveraged text-guided synthetic feature generation for known classes, while employing a novel K-means clustering strategy (K'=K) on high-entropy features for unknown modeling. Specifically, after identifying samples with high entropy, we optimized these features to maximize their entropy further. We then applied K-means clustering to these optimized high-entropy features, creating K' distinct unknown prototype clusters. This approach allows us to model the unknown space with multiple centroids rather than

Table 8: HOS (%) results with foundation model integration on Office-Home.

| Method | Product | RealWorld | Art | Clipart | Average |
|---|---|---|---|---|---|
| CLIP | 50.4 | 56.0 | 62.8 | 61.0 | 57.6 |
| CLIP + RRDA | **70.7** | **73.4** | **74.1** | **65.7** | **71.0** |

a single homogeneous "unknown" class, enabling more nuanced decision boundaries and better separation between known and unknown samples.

Table 8 presents results across different domains, showing that RRDA significantly enhances CLIP's performance for open-set domain adaptation. While zero-shot CLIP provides a strong baseline (57.6% average HOS), RRDA integration improves performance by 13.4 percentage points (71.0% average HOS).

## C Future Directions and Broader Impact

RRDA offers a robust framework for SF-OSDA with several promising directions for future enhancement. Our current implementation uses a fixed relationship between known and unknown classes (K'=K) for simplicity, which has proven effective across diverse benchmarks. Future work could explore adaptive strategies for determining the optimal K' based on target domain characteristics. The performance depends on the quality of synthetic features, where optimizing feature generation techniques could further improve adaptation results.

From a broader perspective, RRDA advances privacy-preserving machine learning by enabling effective adaptation without source data access. This has significant implications for sensitive domains like healthcare and industrial systems, where data sharing is restricted by privacy regulations or proprietary concerns. By improving classification of unknown classes, RRDA also enhances the safety and reliability of deployed systems in open-world environments where novel classes may appear unexpectedly. Additional promising research directions include extending RRDA to continual learning scenarios where new unknown classes appear over time, and exploring its application in multi-modal settings to further extend its practical impact across diverse application domains.

