# OpenReview forum: "Recall and Refine: A Simple but Effective Source-free Open- set Domain Adaptation Framework"
_TMLR — Accepted by TMLR_

### Review · Reviewer_CXK8 · 2025-02-08

**Summary Of Contributions:**

This paper focuses on the challe of source-free and open-set Domain Adaptation (SF-OSDA), where a  pre-trained model from source domain (labeled data are accessible) needs to be adapted to the target domain data that have both unlabeled and new classes not present in the source.

The authors introduce a two-step framework called Recall and Refine (RRDA) designed specifically to overcome the above challenges.

In the first stage, the authors generate synthetic feature points for both known and unknown classes by using the initialized target domain features. For known classes, the optimization is similar to common DA methods, i.e.  minimizing the cross-entropy loss with respect to the source classifier’s prediction. For novel classes, the optimization is guided with a linear combination of entropy and hinge loss, aiming to find out feature points that has entropy greater than a threshold value while avoiding collapse into a degenerated point.

In the second stage, a new classifier is trained that includes decision boundaries for both the K known classes and the K′ unknown classes. The weights corresponding to the known classes are initialized from the source classifier and the authors use loss functions of the off-the-shelf source-free domain adaptation method (such as SHOT or AaD).

The framework is evaluated on standard benchmark datasets such as Office-Home, Office-31, and VisDA.

**Audience:**

Yes

**Broader Impact Concerns:**

No significant concerns and no ethics review needed..

**Claims And Evidence:**

No

**Requested Changes:**

The presentation of the methods part needs to be polished. The analysis of the choice of threshold and computation should be provided.

For a detailed review, please see the Weaknesses.

**Strengths And Weaknesses:**

### **Strength**
- RRDA works in a source-free setting, i.e., it does not require access to the original labeled source data during adaptation. This is especially beneficial when source data cannot be shared due to privacy or legal constraints.

- RRDA explicitly generates synthetic features for both known and unknown classes, which provides potential for better interpretability and differs from conventional methods that rely solely on thresholding prediction entropy to separate known and unknown classes.

- The framework is designed to be compatible with any off-the-shelf source-free domain adaptation method (e.g., SHOT, AaD).

- The framework has been validated on multiple benchmark datasets with comprehensive experiments.

### **Weakness**
- The paper introduces ambiguous notation and definitions, e.g., the labels are represented with both $I_k$ and $y_k$, making the presentation difficult to understand. Some details are also not clearly elaborated. For example, in the sentence "we initialize the optimization with a noisy version of the original features $z_t$", the authors need to clarify what is a noisy version. What noise is added to $z_t$. The original features may point to the features extracted by the source-pre-trained encoder without any optimization, but is not explained here.

- The variance computed in (1) and (2) is not clear. Do authors compute within mini-batches or within the full target dataset? The results could be different and may have different effects.

- The design of the method seems to be heuristic, while the derivation and the rationale behind the choice of the entropy-based loss functions and the accompanying variance regularization terms are not fully explained.

- The optimization in terms of the feature generation is heavily affected by the choice of the threshold, which is jointly decided by the hyperparameter and the number of source classes. The ablation shown in Table 5 is not straightforward to draw the conclusion. The authors should put more discussions on this.

- As RRDA employs a two-stage training process, the authors need to provide an analysis of how synthetic feature generation and the follow-up process increase both computational complexity and training time compared to simpler adaptation methods.

---

> ### Author Response · Authors · 2025-03-20
>
> Dear Reviewer,
>
> Thank you for your thoughtful feedback. We have carefully addressed each concern as follows:
>
> **A1:** We acknowledge that our notation was not sufficiently clear in distinguishing between different representations. In the revised manuscript, we will improve the clarity by consistently using I_k to denote the identity function (one-hot encoding) for the k-th class during synthetic point optimization and y^*t to represent the class labels assigned to synthetic points for target classifier training. Regarding the "noisy initialization" mentioned in the paper, we'll clarify that we add Gaussian noise (*μ = 0, σ*=1.0 )to the features extracted by the source-pretrained encoder. We will also explicitly define what we mean by "original features" - these are the features extracted by the source-pretrained encoder (h_θ^s) before any optimization takes place. These clarifications will make our methodology more transparent and easier to follow.
>
> **A2:** Thank you for this important question. The variance term in Eqs. (1)-(2) is computed **across the full set of synthetic features**, not mini-batches. We will make this explicit in the revised manuscript. Computing variance over the full optimization set rather than mini-batches ensures that we maintain global diversity across all synthetic features. This is particularly important for unknown class generation, where diversity helps represent the range of potential novel classes in the target domain. For more intuition and quantitative results, please see answer A3.
>
> **A3:**  Our entropy-based approach is grounded in key principles from domain adaptation and open-set recognition literature. For known classes, we optimize features to exhibit low entropy with respect to specific classes aligning with SHOT, ensuring that synthetic features match source class centroids.  For unknown classes, we deliberately maximize entropy to identify points near decision boundaries, following [1]. The variance regularization term is inspired by VICReg [2] to prevent feature collapse during optimization, ensuring diversity in our synthetic samples.
>
> We ran the experiment on the entire Office-Home dataset, which is more challenging to see thee impact of the variance term on the performance. We reported the results using SHOT+RRDA.
>
> | Task | A→C | A→P | A→R | C→A | C→P | C→R | P→A | P→C | P→R | R→A | R→C | R→P | Average |
> | --- | --- | --- | --- | --- | --- | --- | --- | --- | --- | --- | --- | --- | --- |
> | Without  the variance term | 59.9 | 70.1 | 68.3 | 60.6 | 67.5 | 68.0 | 58.8 | 58.4 | 75.7 | 56.5 | 55.9 | 77.2 | 64.7 |
> | With the variance term | 64.6 | 74.2 | 77.2 | 63.1 | 71.4 | 71.3 | 67.7 | 59.1 | 76.7 | 70.2 | 67.4 | 76.7 | 70.0 |
> | Improvement | +4.7 | +4.1 | +8.9 | +2.5 | +3.9 | +3.3 | +8.9 | +0.7 | +1.0 | +13.7 | +11.5 | -0.5 | +5.3 |
>
> ---
>
> We observe that the variance term is more important for the difficult transfer tasks such as R→A, R→C .
>
> **A4**:  Thank you for raising this important concern. We conducted a comprehensive analysis across the entire Office-31 tasks using SHOT+RRDA with different thresholds:
>
> | Threshold | H-Score | KnownAcc | UnknownAcc |
> | --- | --- | --- | --- |
> | 0.10/0.9 | 0.705±0.107 | 0.887±0.066 | 0.596±0.133 |
> | 0.20/0.8 | 0.715±0.100 | 0.894±0.070 | 0.611±0.138 |
> | 0.25/0.75 | **0.726±0.098** | **0.900±0.072** | **0.623±0.140** |
> | 0.30/0.7 | 0.720±0.093 | 0.899±0.058 | 0.614±0.132 |
> | 0.40/0.6 | 0.723±0.092 | **0.900±0.061** | 0.616±0.131 |
> | 0.50/0.5 | 0.721±0.092 | 0.890±0.075 | 0.619±0.130 |
>
> These results demonstrate that our method maintains consistent performance across most threshold values (0.2-0.5), with 0.25/0.75 yielding the best overall results. In the revised paper, we will add this detailed analysis and explain how threshold selection offers a trade-off between known-class accuracy and unknown-class discovery. We will also include a similar analysis for AaD+RRDA.
>
> **A5**:  We appreciate your concern about the computational overhead introduced by our two-stage process. To address this, we measured runtime comparisons on the Office-31 A2D task:
>
> | Method | Runtime |
> | --- | --- |
> | LEAD | 3m59s |
> | SHOT | 4m00s |
> | SHOT+RRDA | 4m31s |
> | AaD | 4m01s |
> | AaD+RRDA | 4m35s |
>
> As shown, RRDA introduces only a small computational overhead  (~31 seconds, or approximately 13% increase) while achieving significant performance gains (+5.6% HOS improvement for SHOT+RRDA over SHOT-O on Office-31).  The additional time comes primarily from the synthetic data generation and target classifier training stages, which require only a small fraction of the total adaptation time. We will include this experiment in the revised version of the manuscript.
>
> We hope these clarifications address your concerns. Thank you again for your valuable feedback.

---

> > ### Author Response · Authors · 2025-03-20
> >
> > [1] Saito, K., Yamamoto, S., Ushiku, Y., & Harada, T. (2018). Open set domain adaptation by backpropagation. In *Proceedings of the European conference on computer vision (ECCV)* (pp. 153-168).
> >
> > [2] Bardes, A., Ponce, J., & Lecun, Y. (2022, April). VICReg: Variance-Invariance-Covariance Regularization For Self-Supervised Learning. In *ICLR 2022-International Conference on Learning Representations*.

---

> > > ### Comment · Reviewer_CXK8 · 2025-04-03
> > >
> > > I acknowledge I have read the rebuttal and other views.The authors have provided comprehensive studies and thorough revision and I appreciate their efforts. My concerns have been addressed.

---

> > > > ### Author Response · Authors · 2025-04-04
> > > >
> > > > We are glad to hear that we have addressed your concerns! Thanks for spending a significant amount of time on our submission and giving lots of insightful feedback, which significantly strengthen our paper! We will include all added experiments and points in the final paper for better clarification.

---

### Review · Reviewer_rcMJ · 2025-02-27

**Summary Of Contributions:**

1. Introduction of RRDA Framework: The paper presents Recall and Refine (RRDA), a novel framework for Source-free Open-set Domain Adaptation (SF-OSDA) that explicitly learns discriminative features for target-private unknown classes, addressing significant limitations in existing methods.

2. Two-Step Process: RRDA employs a two-step approach to enhance the model's ability to recognize unknown classes. The first step involves training a target classifier with an additional decision boundary using synthetic samples from target domain features. The second step adapts the entire model to the target domain, addressing domain shifts and improving generalization to unknown classes.

3. Integration with Existing Methods: The framework allows for the seamless integration of various off-the-shelf source-free domain adaptation methods, such as SHOT and AaD, enhancing its applicability across different scenarios.

4. Empirical Validation: Comprehensive experiments conducted on three benchmark datasets demonstrate that RRDA significantly outperforms existing SF-OSDA and OSDA methods, providing strong evidence of its effectiveness.

**Audience:**

Yes

**Claims And Evidence:**

No

**Requested Changes:**

See ***Weaknesses***.

**Strengths And Weaknesses:**

***Strengths***

1. Novel Approach: The RRDA framework addresses critical gaps in current SF-OSDA methods by focusing on the explicit learning of features for unknown classes, thereby enhancing the model's recognition capabilities.

2. Effective Decision Boundary: The introduction of an additional decision boundary in the target classifier effectively separates known and unknown classes, improving classification accuracy.

3. Flexibility and Applicability: The ability to integrate with various existing source-free domain adaptation methods increases the versatility and practical relevance of the framework in real-world applications, particularly under privacy constraints.

4. Robust Experimental Results: The extensive empirical validation on benchmark datasets solidifies the framework's credibility and establishes a new standard for performance in the field of SF-OSDA.


***Weaknesses***

1. How does SF-OSDA differ from traditional out-of-distribution (OOD) detection and open-set domain adaptation (DA) in terms of problem definition, task characteristics, significance, challenges, and real-world applicability, particularly considering that open-set DA also contends with distribution shifts and novel classes? Should state-of-the-art methods from both branches be compared with the proposed framework? Additionally, what evidence supports the claim that the proposed framework can outperform existing approaches?

2. I recall that some open-set domain adaptation studies have explicitly focused on learning features for target-private unknown classes. Additionally, the proposed RRDA appears to be a straightforward combination of OOD detection and SFDA, raising concerns about its technical novelty.The authors may provide further clarification.

3. How should the appended unknown-class target classifier with K' neurons be initialized? How does the random initialization differ from using source-pretrained weights?

4. Figure 1 effectively illustrates the effects and mechanisms of RRDA; however, it lacks a clear depiction of the workflow and network structure, which are crucial for understanding the technical implementation.

5. High entropy with respect to the source classifier does not necessarily indicate unknown classes; it may also reflect hard or uncertain samples from known classes. Given this insight, it may be inappropriate to label this process as "Synthetic Unknown Classes Generation".

6. In Eq. (1), what is the underlying principle that dictates the standard deviation of the features of a synthesized sample must be equal to or greater than 1?

7. Due to the thresholding in Synthetic Unknown Classes Generation and Synthetic Known Classes Generation, class imbalance may arise. How does this imbalance affect the final adaptation process? As a potential remedy, could z^t be sampled from a Gaussian distribution and then optimized for high/low entropy and high variance? Additionally, can we design a neural network generator to translate an input z^t to a desired z^{*t} more effectively than through ontology optimization?

8. The authors could enhance their validation by extending it to larger datasets, such as S2RDA [a], and comparing their results to the benchmarks established in that study.

[a] A New Benchmark: On the Utility of Synthetic Data with Blender for Bare Supervised Learning and Downstream Domain Adaptation. CVPR, 2023.

9. In Table 4, it appears that diversity has minimal impact on the results. Why?

---

> ### Author Response · Authors · 2025-03-20
>
> Dear Reviewer,
>
> Thank you for your thorough and insightful feedback. We appreciate the opportunity to clarify and improve our work. Below, we address each of your concerns in detail:
>
> **A1:** Source-Free Open-Set Domain Adaptation (SF-OSDA) fundamentally differs from both traditional Out-of-Distribution (OOD) detection and Open-Set Domain Adaptation (DA) in problem definition and task characteristics. While OOD detection focuses solely on identifying unknown classes **without adapting the model**, and Open-Set DA assumes **access to source data during adaptation**, SF-OSDA operates under the stricter constraint of **no access to source data**, addressing both domain shift and novel class discovery simultaneously. This makes SF-OSDA particularly relevant for real-world applications with privacy constraints (e.g., industrial systems, especially fault detections in PHM tasks), where source data cannot be shared.
>
> In our experimental evaluation, we comprehensively compared RRDA against both state-of-the-art open-set DA methods (OSBP, CMU, STA, ANNA) and universal source-free DA approaches (UMAD, LEAD, GLC), following established protocols from previous studies.
>
> A2: To the best of our knowledge, existing SF-OSDA methods primarily rely on thresholding entropy or confidence scores to identify unknown classes rather than explicitly learning their semantics. Our approach uniquely generates synthetic features for both known and unknown classes, allowing direct learning of decision boundaries for novel categories without source data access.
> If there are any suggested methods you would like us to compare against in open-set scenarios, we would be happy to include them in a revised version of our manuscript.
>
> A3:  The target classifier is initialized as follows:
>
> - **Known Classes (*K*)**: Initialized with source classifier weights to preserve source knowledge.
> - **Unknown Classes (*K*′)**: Randomly initialized, as the source model has never seen the unknown classes, hence, there is no decision boundary learned for them.
>
> A4: We acknowledge the lack of clarity in Figure 1, we will improve the figure to include clearer step-by-step textual descriptions of the process flow, making the technical implementation of RRDA more accessible to readers.
>
> A5:  Thank you for this insightful question. You highlight an important distinction - if we simply consider actual target samples with high entropy as unknown, we would encounter the same limitations as existing thresholding methods.
> Our approach differs fundamentally: the optimized high-entropy points are not actual target domain samples but synthetic features specifically designed to represent unknown class characteristics. These synthetic points are used only to train the target classifier, establishing decision boundaries for unknown classes. During adaptation, real target samples are classified based on these learned boundaries, not by entropy thresholding.
>
> A6: The constraint Var(z^t)≥1 ensures **feature diversity** during optimization, preventing the generated points from collapsing to a single location in feature space. This approach was inspired by VICReg [1], which similarly maintains representation variance to avoid collapse in self-supervised learning.
>
> Our experiments demonstrate the importance of this constraint. Without variance regularization, performance on Office-Home drops from 70.0% to 64.7% average HOS (-5.3%), with particularly significant degradation on challenging tasks like R→A (-13.7%). The full table is in answer 9.
>
> A7: Thank you for raising this point.  In our implementation, we address this by:
>
> 1. Capping the number of samples per class to ensure balance (1000 for Office-Home/Office-31, 10,000 for VisDA)
> 2. Using a fixed K' = K for simplicity and balance
>
> Regarding alternative approaches, sampling from a Gaussian distribution and then optimizing is an interesting direction. However, our experiments showed that starting from actual target features provides better initialization for optimization. A neural network generator is indeed a promising alternative that could potentially improve efficiency. We view these suggestions as valuable future work directions that could build upon our current framework.
>
> A8: Thank you for suggesting the S2RDA benchmark. In our current work, we focused on the standard benchmarks used in OSDA literature (Office-31, Office-Home, VisDA) to enable direct comparison with existing methods. The synthetic nature of S2RDA is particularly relevant to our approach, which also leverages synthetic data (albeit in feature space rather than image space). We agree this would be a valuable direction for future work and will discuss it in the revised manuscript.

---

> > ### Author Response · Authors · 2025-03-20
> >
> > A9: We conducted additional experiments to more thoroughly evaluate the impact of diversity regularization on the challenging Office-Home dataset:
> >
> > | Task | A→C | A→P | A→R | C→A | C→P | C→R | P→A | P→C | P→R | R→A | R→C | R→P | Average |
> > | --- | --- | --- | --- | --- | --- | --- | --- | --- | --- | --- | --- | --- | --- |
> > | Without  the variance term | 59.9 | 70.1 | 68.3 | 60.6 | 67.5 | 68.0 | 58.8 | 58.4 | 75.7 | 56.5 | 55.9 | 77.2 | 64.7 |
> > | With the variance term | 64.6 | 74.2 | 77.2 | 63.1 | 71.4 | 71.3 | 67.7 | 59.1 | 76.7 | 70.2 | 67.4 | 76.7 | 70.0 |
> > | Improvement | +4.7 | +4.1 | +8.9 | +2.5 | +3.9 | +3.3 | +8.9 | +0.7 | +1.0 | +13.7 | +11.5 | -0.5 | +5.3 |
> >
> > These results demonstrate that diversity regularization has a more significant impact on challenging datasets with complex domain shifts. The improvement is particularly pronounced on difficult transfer tasks like R→A (+13.7%) and R→C (+11.5%).
> >
> > We sincerely thank you for your rigorous review, which has significantly strengthened our work. All revisions will be incorporated into the manuscript, and we remain open to further suggestions.
> >
> > [1] Bardes, A., Ponce, J., & Lecun, Y. (2022, April). VICReg: Variance-Invariance-Covariance Regularization For Self-Supervised Learning. In *ICLR 2022-International Conference on Learning Representations*.

---

> > ### Comment · Reviewer_rcMJ · 2025-03-31
> >
> > The response has effectively addressed my concerns. The authors should revise the manuscript in accordance with the feedback provided, specifically by clarifying the problem definition, enhancing the clarity of Figure 1, and including the S2RDA benchmark.

---

> > > ### Author Response · Authors · 2025-04-01
> > >
> > > Thank you for your valuable feedback! We're pleased to hear that our response has addressed your concerns. We will revise the manuscript according to your suggestions. We appreciate your thoughtful input, which has helped strengthen our paper significantly.

---

### Review · Reviewer_jPFV · 2025-02-28

**Summary Of Contributions:**

This paper tackles the source-free open-set domain adaptation (OSDA) problem and proposes a novel method, PRDA, which introduces synthetic features for both known and unknown classes. The proposed PRDA method effectively integrates existing source-free OSDA approaches, and the experimental results demonstrate a significant improvement in performance.

**Audience:**

Yes

**Broader Impact Concerns:**

Not available

**Claims And Evidence:**

Yes

**Requested Changes:**

Exploration of Foundation Models: Given the advancements in modern foundation models, it is recommended to investigate how the proposed method could be extended or adapted to work with models such as DINOv2 or CLIP. This would enhance the method's applicability and relevance in the context of current trends in the field.

Addressing Varying Ratios of Unknown Samples: The issue of varying ratios of unknown samples across different practical scenarios is a critical challenge in source-free OSDA. It is suggested that the authors discuss potential strategies or insights to address this limitation, as it is a common weakness shared by existing methods in this domain.

Additionally, a deeper analysis of the limitations and potential broader impacts of the method would further enhance the paper's rigor and completeness. Overall, this work offers a meaningful contribution to the field of source-free OSDA.

**Strengths And Weaknesses:**

## Strengths
-  The introduction of synthetic samples in the feature space, coupled with a learnable method for generating new samples, is a novel approach and can be regarded as a significant contribution to the field of source-free OSDA.

- Additionally, the experiments are comprehensive and sufficiently demonstrate the effectiveness of the proposed method across several conventional benchmarks.

## Weaknesses
- With the rapid development of modern foundation models, conventional model adaptation methods may lack broader insights, as they are often sensitive to data characteristics. It would be interesting to explore how the proposed method could be adapted or applied to foundation models such as DINOv2 or CLIP.
- The practical challenge of varying ratios of unknown samples across different scenarios remains unaddressed, which is a common limitation shared by existing source-free OSDA methods.

---

> ### Author Response · Authors · 2025-03-20
>
> Dear Reviewer,
>
> Thank you for your valuable feedback and constructive suggestions. We address your points below and will incorporate these improvements into the revised manuscript:
>
> A1:  We agree that integrating foundation models (e.g., DINOv2, CLIP) could enhance RRDA's applicability. While foundation models excel in general-purpose tasks, their effectiveness in **specialized domains** (e.g., industrial Prognostics and Health Management) is often limited due to domain specificity.     To ensure broad usability, RRDA was designed with compatibility for standard models (e.g., ResNet, ViT) while retaining the flexibility to adapt to newer architectures. As a preliminary validation, we systematically evaluated how foundation models like CLIP impact our method compared to our original ViT implementation. Our experiments revealed two critical insights. First, regarding architecture parity, when fine-tuned using our framework, CLIP's image encoder achieved comparable performance to our originally reported ViT results.
>
> Second, in terms of zero-shot advantage, CLIP's pre-trained cross-modal capabilities enabled superior zero-shot performance compared to standard CLIP's.
>
> For our experimental setup, we used the Office-Home dataset (65 classes) with Clipart (C) as the target domain. We didn't finetune the model, and considered 25 shared classes. Our CLIP configuration included both zero-shot classification using text embeddings of "a photo of [CLASS]" prompts, and our method initialized with CLIP-ViT-B/32 image encoder. We augmented this with entropy-based unknown detection (threshold = 0.5 × log(25)), as we considered only the known classes as prompts.
>
> For our approach, we leveraged text-guided synthetic feature generation for known classes, while employing a novel K-means clustering strategy (K'=K) on high-entropy features for unknown modeling. Specifically, after identifying samples with high entropy, we optimized these features to maximize their entropy further. We then applied K-means clustering to these optimized high-entropy features, creating K’ distinct unknown prototype clusters. This approach allows us to model the unknown space with multiple centroids rather than a single homogeneous "unknown" class, enabling more nuanced decision boundaries and better separation between known and unknown samples.
>
> The results demonstrate the effectiveness of our approach across different domains:
>
> |  | Product | RealWorld | Art | Clipart | Average |
> | --- | --- | --- | --- | --- | --- |
> | Clip | 50.4 | 56.0 | 62.8 | 61.0 | 57.6 |
> | Clip + ours | 70.7 | 73.4 | 74.1 | 65.7 | 71.0 |
>
> A2:     We thank the reviewer for highlighting this critical challenge. As shown in Figure 3(a), RRDA already exhibits robustness to varying openness levels, consistently outperforming baselines like LEAD and SHOT-O.
>
> To further enhance adaptability, we propose two strategies:
>
> (1) dynamically adjusting the number of unknown clusters *K*′ based on the estimated unknown ratio during synthetic feature generation (instead of assuming K’=K), and (2) rebalancing loss weights in Eq. (3) using pseudo-label confidence scores to mitigate class imbalance.
>
> A3:      Our work has three primary limitations: (1) RRDA’s performance depends on the quality of synthetic features, where poor optimization may propagate errors during adaptation; (2) the assumption of a fixed K′=K for unknown clusters may not perfectly align with real-world class distributions; and (3) the method introduces a small computational overhead.
>
> Despite these limitations, RRDA advances privacy-preserving machine learning by enabling adaptation without source data access. This has implications for sensitive domains like healthcare and industrial systems, where data sharing is restricted by privacy regulations or proprietary concerns
>
> We sincerely appreciate your constructive feedback, which has strengthened our work. The revised manuscript will incorporate these improvements, including expanded discussions on the foundation model. Thank you again for your rigorous review and valuable insights.

---

### Decision · Action_Editor_4XDT · 2025-04-18

**Recommendation:** Accept as is

**Comment:**

All reviewers agree that the current version can be accepted. Thus, no further revision is required at this stage.

**Audience:**

This paper fits the TMLR's audience, as the problem studied is classical in the field of domain adaptation.

**Claims And Evidence:**

This paper's claims are supported by the evidence very well, although the initial submission has some issues regarding the presentation. After the author-reviewer responses, the presentation issue and other issues are addressed. Thus, I recommend accepting this paper.